# Mutational Landscape and Precision Medicine in Hepatocellular Carcinoma

**DOI:** 10.3390/cancers15174221

**Published:** 2023-08-23

**Authors:** Leva Gorji, Zachary J. Brown, Timothy M. Pawlik

**Affiliations:** 1Department of Surgery, Kettering Health Dayton, Dayton, OH 45405, USA; leva.gorji@ketteringhealth.org; 2Department of Surgery, Division of Surgical Oncology, New York University—Long Island, Mineola, NY 11501, USA; zachary.brown2@nyulangone.org; 3Department of Surgery, Division of Surgical Oncology, The Ohio State University Wexner Medical Center and James Cancer Hospital, Columbus, OH 43210, USA

**Keywords:** precision medicine, hepatocellular carcinoma, targeted therapy

## Abstract

**Simple Summary:**

Hepatocellular carcinoma (HCC) continues to manifest a global burden with increasing incidence. Although many treatment modalities exist, the disease process is plagued with high recurrence and mortality. The comprehension of the mutation landscape of the disease in conjunction with the use of precision medicine will aid in individualized, targeted therapies for treatment. Through understanding the tumor microenvironment, management of the malignancy may be optimized. The purpose of our review is to emphasize the role and prospects of precision medicine in the treatment of HCC.

**Abstract:**

Hepatocellular carcinoma (HCC) is the fourth most common malignancy worldwide and exhibits a universal burden as the incidence of the disease continues to rise. In addition to curative-intent therapies such as liver resection and transplantation, locoregional and systemic therapy options also exist. However, existing treatments carry a dismal prognosis, often plagued with high recurrence and mortality. For this reason, understanding the tumor microenvironment and mutational pathophysiology has become the center of investigation for disease control. The use of precision medicine and genetic analysis can supplement current treatment modalities to promote individualized management of HCC. In the search for personalized medicine, tools such as next-generation sequencing have been used to identify unique tumor mutations and improve targeted therapies. Furthermore, investigations are underway for specific HCC biomarkers to augment the diagnosis of malignancy, the prediction of whether the tumor environment is amenable to available therapies, the surveillance of treatment response, the monitoring for disease recurrence, and even the identification of novel therapeutic opportunities. Understanding the mutational landscape and biomarkers of the disease is imperative for tailored management of the malignancy. In this review, we summarize the molecular targets of HCC and discuss the current role of precision medicine in the treatment of HCC.

## 1. Introduction

Hepatocellular carcinoma (HCC) is the fourth most common malignancy globally, with an incidence of nearly 900,000 cases annually [1,2]. HCC constitutes the majority of primary liver cancers and accounts for nearly 5% of all malignant neoplasms. As the tumor can initially be clinically indolent, diagnosis is typically made at an advanced stage of the disease. The most common risk factors for the development of HCC include viral hepatitis, alcohol-induced liver disease, non-alcoholic fatty liver disease (NAFLD), and non-alcoholic steatohepatitis (NASH) (Figure 1) [3,4,5,6,7]. Notably, virally associated HCC is attributable to the development of specific encoded viral proteins. However, tumor carcinogenesis differs, even amongst viral etiologies; for example, hepatitis B virus (HBV) may integrate into the host chromosome, while hepatitis C virus (HCV) is unable to do so. Additionally, while persistent inflammation, immune dysregulation, and anomalous lipid breakdown contribute to the development of HCC in viral etiologies and NAFLD, the epigenetics differ. While DNA methylation is important, in NASH-induced disease, methylation results in gene silencing of DNA methyltransferase, which is associated with progressive fibrosis, lipid and glucose metabolism, and DNA repair; in HBV-related HCC, silencing of tumor suppressor genes occurs with the methylation of CpG islands of these genes. In addition, for HCV-related HCC, DNA methylation may occur at the Gadd45B, which results in ineffective cell cycle arrest and tumorigenesis [8,9,10,11].

HCC treatment is complex and must take into account tumor burden, liver function, and the patient’s physical status to identify the appropriate treatment paradigm, which may include ablation, resection, liver transplant (LT), locoregional, or systemic therapies [12]. Over the past several years, there has been improvement in systemic therapies for patients with HCC, but response rates and survival remain poor. The development of personalized therapeutic strategies utilizing precision medicine (PM) may demonstrate the ability to improve outcomes. In current practice, there are no “perfect” biomarkers available to diagnose malignancy, nor are there biomarkers available to predict prognosis with high accuracy [13].

PM is a therapeutic paradigm that attempts to implement the optimal treatment strategy for individualized disease process characteristics. The PM approach accounts for the heterogeneity of the disease process and individual patient characteristics, seeking to tailor and personalize oncologic treatment [14]. Emerging technologies have allowed for serial monitoring of disease treatment response and progression through molecular analysis of liquid samples (Table 1) [15,16,17,18,19,20,21,22,23]. In this review, we discuss the molecular landscape of HCC and the use of PM to assist in determining prognosis, diagnosis, and guidance of therapy. Specifically, a systematic review of PubMed and Embase between 2003 and 2023 was conducted using the following MESH terms: precision medicine, HCC, genomic mutations, and targeted therapies.

## 2. Mutational Landscape of HCC

Tumorigenesis may occur because of cell transformation from proto-oncogene or tumor suppressor gene mutations. The most common mutations noted in HCC include TERT, TP53, CTNNB1, AXIN1, ARID1A, CDKN2A, and CCND1 genes (Table 2) [24,25,26,27,28,29,30,31,32,33,34,35,36,37,38,39,40,41,42,43,44]. Many common HCC mutations are not actionable, and therapies with proven clinical benefit are still lacking. Therefore, treatment outside of clinical protocols is largely not recommended.

### 2.1. Microsatellite Instability

Microsatellite instability (MSI) is caused by erroneous mismatch repair, which leads to the accumulation of DNA microsatellites. There are multiple ways in which deficient mismatch repair (dMMR) may occur, including somatic mutations, hypermethylation of MMR protein genes, and miRNA-mediated downregulation [45]. MSI-high (MSI-H) accounts for a small percentage of HCC tumors, occurring in up to 2.9% of HCC lesions [46,47,48]. Muakai et al. assessed the prevalence of MSI-H in 50 patients with HCC and identified one patient with the mutation pattern. Although the tumor did not demonstrate PD-L1 expression, it did exhibit shrinkage with the administration of pembrolizumab [45].

### 2.2. BRCA and BRCAness Mutations

Breast cancer gene 1 (BRCA1) is a tumor suppressor gene with a known increased risk of breast cancer. Mei et al. examined the relationship between the BRCA1 mutation and HCC and determined a correlation between BRCA1 and advanced T stage, clinical stage, poor tumor grade, and MSI status, as well as reduced recurrence-free survival (RFS) and reduced overall survival (OS) [49].

### 2.3. Gene Fusions

Gene fusions play a vital role in the initial steps of tumorigenesis and occur as a result of the transcription and translocation of two genes as a single unit. Investigations of gene fusions have been pursued as potential therapeutic targets. Several known gene fusions exist for HCC, including MAN2A1-FE R and DNAJB1-PRKACA. In vivo and in vitro studies have demonstrated greater sensitivity to FER inhibitor crizotinib and EGFR inhibitor canertinib among patients who possess the MAN2A1-FER gene fusion [50,51]. Notably, the DNAJB1-PRKACA fusion gene is present in virtually all patients with the fibrolamellar variant of HCC. This gene fusion results in the enhanced catalytic effect of protein kinase A and subsequent tumorigenesis. Toyota et al. evaluated the impact of a novel PRKACA inhibitor, DS89002333, on the fibrolamellar variant of HCC. Both in vivo and in vitro data demonstrated inhibitory activity and inhibited fusion protein-dependent cell growth [52]. However, due to the rarity of this variant, research on targeted therapies has been difficult.

### 2.4. Omics Signature

Transcriptomics and proteomics signatures, referred to as omics signatures, are high-dimensional molecular-level measurements that enable early HCC detection and prediction of prognosis. The omics signature is a readout of the changes in gene and protein expression levels after perturbation [53]. Wu et al. proposed a procedure for multi-omics gene pair signature identification using methylome and transcriptome data as potential molecular targets for HCC [54].

### 2.5. Mutational Burden

The tumor mutational burden (TMB) has also been a focus of investigation in which the total number of mutations per coding area of a tumor genome has been explored [55]. Xie et al. investigated the potential of TMB as a prognostic indicator among patients with HCC. Using multiple databases and whole exome sequencing, *t* high TMB was noted to be associated with a worse prognosis and higher risk of relapse compared with low TMB HCC [56]. Therefore, to predict prognosis, further investigation should be pursued to optimize the role of TMB to individualize care and PM.

### 2.6. TERT

Tumorigenesis is driven by a unique combination of somatic mutations [57]. The most common mutation in HCC is the reactivation of TERT, which encodes for a rate-limiting catalytic subunit of telomerase [58]. An aberrant TERT mutation exists in 95% of solid tumors [58]. Nearly 60% of TERT mutations occur in the promoter region. These TERT promoter (TERTp) mutations are the earliest somatic alterations in HCC and create de novo binding sites for the E-twenty-six (ETS) transcription factors and often increase TERT expression [59,60]. The telomere-related functions are referred to as the canonical role of TERT, but TERT also has non-canonical roles, which include the regulation of metabolic mechanisms, stress responses, RNA splicing, and involvement of signal transduction pathways such as the Wnt pathway and the c-MYC pathway [59,61,62].

Zhou et al. reported on TERT mutation as it pertains to HCC using NGS. The mutant TERTp group demonstrated upregulation of 536 IncRNAs, 21 circRNAs, 41 miRNAs, and 266 mRNAs and downregulation of 1745 IncRNAs, 23 circRNAs, 32 miRNAs, and 1117 mRNAs (*p* < 0.05) versus the wild-type TERTp. Zhou and colleagues developed a differentially expressed IncRNA/circRNA-miRNA-mRNA network to depict the effects of the mutation on ncRNA regulation. Subsequently, the carcinogenic probability was identified in two ncRNA regulatory axes, potentially contributing to HCC progression [63]. Due to the increasing prevalence of HCC in Asian countries, Trung et al. examined the clinical significance of the TERTp mutation among patients with HCC diagnosis in Eastern countries. The data suggested monitoring levels of TERTp mutations (C228T and C250T), miRNA-122, and AFP levels as potential biomarkers to assist with the diagnosis of HBV-induced HCC in the Eastern population of patients [64]. As the role of a mutation in TERTp continues to be investigated in HCC, further targeted therapies and clinical applications will be defined.

### 2.7. PT53

The second most common mutation in HCC, identified in 30% of cases, involves the TP53 tumor suppressor gene [51]. Following damage induced to DNA, wild-type TP53 has a significant role in apoptosis, preventing dysregulation of cellular proliferation. Mutant TP53 proteins lose tumor suppression and promote tumorigenesis [65]. Mutations of the TP53 gene in HCC have been associated with a poor prognosis, particularly when associated with hotspot mutations R249S and V157F [66]. Patients with TP53 mutations often demonstrate decreased recurrence-free survival (RFS) and overall survival (OS) [67]. By identifying a gene signature from the transcriptome of HCC patients functionally related to mitotic cell cycle regulation, Lin et al. demonstrated the sensitivity of HCC lines with mutant TP53 or wild-type CTNNB1 genes to taxanes using gene–drug association analysis. As a result, the study suggested a benefit of treatment with paclitaxel in this patient population [68]. A poor prognosis-associated signature was identified by Yang et al. in patients with the TP53 mutation; subsequently, in silico screening of three targets (CANT1, CBFB, and PKM) and two agents (irinotecan and YM-155) were noted to have a potential therapeutic impact on patients with a poor prognosis-associated signature [69]. Collectively, the data from the studies mentioned above highlight the potential of personalized risk stratification and precision therapy.

### 2.8. WNT-ß-Catenin

Approximately 20–30% of patients with HCC express a mutation in the WNT-ß-catenin pathway. The Wnt signaling pathway is divided into non-canonical and canonical pathways. The pathway begins with the interaction of the Wnt ligand and the frizzled receptors (FZDs), a seven-transmembrane protein belonging to the G-protein-coupled receptors [70]. Classically, the pathway results in the translocation of ß-catenin into the cell nucleus, where it upregulates transcription of the genes associated with Wnt [70,71]. The activated complex of ß-catenin, TCF, and LEF results in cellular proliferation and survival gene transcription. Aberrant activation of the pathway leads to dysregulated cellular proliferation [72,73].

A proportion of 27% of patients with HCC have gain-of-function mutations of CTNNB1 that code for ß-catenin, which may occur in addition to missense mutations of CTNNB1 that prevent phosphorylation of ß-catenin [72,73]. Mutations of APC and AXIN1 encoding for the degradation complex of ß-catenin may also occur in 3–8% of patients with HCC [72,74]. Currently, a phase 1b clinical trial is being pursued to determine the maximum tolerated dose of OMP-54F28, a recombinant protein that binds to the Wnt ligands functioning as an FZD8 decoy receptor, in combination with sorafenib, among patients with HCC (NCT02069145). Salinomycin is a potassium ionophore that has been demonstrated to inhibit proximal Wnt signaling by interfering with the Wnt coreceptor lipoprotein receptor-related protein 6 and inducing its degradation [75]. Wang et al. investigated the role of salinomycin as a targeted therapy for HCC using three HCC lines: HepG2, SMMC-7721, and BEL-7402. Salinomycin caused cell cycle arrest among the varying HCC lines in different stages of the cell cycle, and ß-catenin expression was downregulated. In vivo analysis of salinomycin using a hepatoma orthotopic tumor model demonstrated HCC size reduction versus the control via inhibition of cell proliferation and induction of apoptosis based on immunohistochemistry and TUNEL staining. Furthermore, in vivo Western blot and IHC demonstrated inhibition of the Wnt/ß-catenin signaling pathway [76]. Using PM to detect aberrations in the Wnt signaling pathway may identify therapeutic targets and act as a prognostic marker for therapeutic response among patients with HCC.

### 2.9. ARID1A

Mutations of the SWI/SNF chromatin remodeling complex subunits are present in approximately 20% of human cancers. The most commonly mutated component of this chromatin remodeling complex is the ARID1A gene, which demonstrates an aberrant variant in 6% of all malignancies, suggesting a broad tumor suppressor function [77]. ARID1A regulates gene transcription, DNA binding, homologous recombination, tumor suppression, DNA damage response, and steroid receptor signaling [78,79]. In a study by Xiao et al., ARID1A knockout resulted in a poor prognosis, was associated with HCC cell growth through high levels of MYC, and demonstrated apoptosis with impaired DNA damage repair subsequent to radiation stress [80]. Abdel-Moety et al. reported similar outcomes in which nuclear expression of ARID1A was markedly lower in HCC compared with surrounding cirrhotic tissues (*p* = 0.002) without a difference in cytoplasmic ARID1A expression. ARID1A nuclear expression was inversely related to tumor size (*p* = 0.006), pathology grade (MCp = 0.046), and post-microwave ablation tumor recurrence (FEp = 0.041) among patients with BCLC stages 0/A eligible for ablation [81].

### 2.10. CDKN2A

Mutation of the CDKN2A gene is involved in tumor advancement through resistance in chemotherapy response, induction of angiogenesis, inhibition of apoptosis, and promotion of cellular proliferation [82,83]. Using The Cancer Genome Atlas (TCGA) and Gene Expression Omnibus (GEO) datasets, Luo et al. evaluated the presence of CDKN2A in HCC and determined that there was a higher risk of developing HCC with increased CDKN2A expression. Furthermore, increased expression was associated with reduced OS and DFS (*p* = 0.003). CDKN2A expression is associated with increases in CD+8 T cells, CD+4 T cells, macrophages, neutrophils, and dendritic cells [82]. These outcomes further support the contribution of immune dysregulation and inflammation to the development of HCC [84,85].

### 2.11. CCND1

The CCND1 gene plays a significant role in the tumorigenesis of multiple malignancies by regulating the G1/S transition in mitosis and activating PAK-2p34 regulation through proteasome-mediated degradation [86,87]. CCND1 plays a relevant role in the sensitivity to chemotherapy agents. Ding et al. examined the role of CCDN1 in chemoresistance to 5′Fleurouracil (5′FU) in HepG2 and SMMC-7721 HCC cell lines. In the study, CCND1 mRNA levels were examined by qRT-PCR, y-H2AX, and RAD51 protein levels determined by Western blot, and CD122+ cell percentage was detected by flow cytometry. CD133 was utilized as a liver cancer stem cell marker, y-H2AX was a marker for DNA damage, and RAD51 was used as a protein marker for DNA repair. CCND1 silencing increased protein levels of y-H2AX, decreased RAD51 expression with the presence of 5′FU, and enhanced the sensitivity of the HepG2 and SMMC-7721 cell lines [88]. These findings warrant further investigation into the role of CCND1 in treating patients with HCC.

### 2.12. Angiogenesis Pathways

The vascular endothelial growth factor (VEGF) is a genomic amplification present in 3–7% of patients with HCC [89]. Elevated levels of VEGF have been associated with decreased OS and progression-free survival (PFS) in patients with HCC [90,91]. VEGF, located on chromosome 6p21, promotes increased tumor vascularity with subsequent tumor growth and dissemination [57,89]. The pathway begins with VEGF (-A, -B, -C, -D, -E) binding with a VEGF receptor (VEGFR), most commonly VEGF-A to VEGFR-2. The ligand-to-receptor binding activates the phosphorylation cascade, resulting in cellular proliferation, chemotaxis of endothelial cells, and increased vascular permeability [92]. 

Zhang et al. highlighted the relevance of Ki67, p53, and VEGF expression in 60 patients with HCC who underwent LT. VEGF was determined to be an independent predictor for HCC recurrence following LT (*p* = 0.005). Additionally, VEGF was associated with the diameter and number of tumors, tumor differentiation, and lymph node metastasis, while p53 was associated with tumor diameter and tumor encapsulation [93]. Lacin et al. suggested that VEGF-A levels be utilized as a serum biomarker to predict treatment response in HCC. In a study of 84 patients, VEGF-A levels correlated with tumor size. Furthermore, VEGF-A levels ≥ 100 pg/mL demonstrated a significant relationship with OS (*p* = 0.01); median OS in patients with VEGF-A levels ≥100 pg/mL vs. <100 pg/mL was 5.8 months versus 14.2 months, respectively (*p* = 0.02) [94]. 

Platelet-derived growth factor (PDGF) is a potent stimulator of angiogenesis and the proliferation of hepatic stellate cells (HSCs). PDGF ligands potentiate activity through tyrosine kinase receptors, consisting of PDGF receptor (PDGFR)-α and -β. An association has been reported of the increased expression of the PDGF ligand A and PDGFR-α with HSC proliferation, liver fibrosis, and HCC tumorigenesis [95,96]. Aryal et al. investigated the role of PDGF-BB. This cytokine exerts a mitogenic impact on hepatic cells, with involvement in malignant transformation, and demonstrated potential clinical significance in serum surveillance following curative resection of HCC. At 2-year follow-up, postoperative serum PDGF-BB < 2133.29 pg/mL was associated with higher HCC recurrence (95% CI, *p* < 0.001) [97].

## 3. Precision Medicine to Guide Therapy

### 3.1. Arterially Directed Therapies

Arterially directed therapies for the treatment of HCC include transarterial embolization (TAE), transarterial chemoembolization (TACE), and transarterial radioembolization (TARE). Tumor and patient characteristics may impact the tumor response to these locoregional therapies, including alpha-fetoprotein (AFP) levels, performance status, neutrophil-to-lymphocyte ratio, and Child–Pugh classification [98]. Arterially directed therapies operate on the premise of targeting the high vascularity that is associated with HCC. However, as 40% of patients demonstrate a minimal response to these interventions, PM has been pursued to investigate resistance [99]. A study by Ziv et al. evaluated the role of nuclear factor E2-related factor (NRF2) in response to TAE/TACE. NRF2 is a component of a subfamily of basic region leucine zipper transcription factors; it appears to mediate drug-metabolizing enzymes and plays a major role in resistance to oxidative stress [100]. Patients with mutations in the NRF2 pathway exhibited rapid tumor progression after TAE/TACE; NRF2-mutated pathways demonstrated a local time to progression of 56% versus 22% among patients without the mutation (*p* < 0.001), implicating ischemia resistance in the setting of overexpression of NRF2. NRF2 knockdown by a short hairpin RNA or NRF2 inhibitor was tested, and there was a synergistic effect related to NRF2 knockdown and ischemia in overexpressing HCC cell lines [101]. Martin et al. sought to investigate the pyruvate kinase muscle isozymes M2 (PKM2) gene as a possible source of resistance to TACE in HCC [102]. PKM2 is a variant of pyruvate kinase and promotes the growth of malignant cells through the Warburg effect [103]. In this study, in vitro TACE models were utilized to measure response to chemotherapy under hypoxia. Knockdown of PKM2 and pharmacologic inhibition of PKM2 resulted in improved drug sensitivity of doxorubicin and cisplatin with TACE [102]. Therefore, an improved understanding of pathways of resistance may lead to improved patient selection and response (Figure 2).

### 3.2. Curative-Intent Interventions

BCLC guidelines recommend radiofrequency ablation (RFA), liver resection (LR), and LT based on the stage of HCC and the patient’s underlying liver condition [12]. The degree of liver cirrhosis, steatosis, and fibrosis, as well as the presence of metabolic syndromes, should be considered when pursuing LR, as these factors contribute to postoperative morbidity and mortality [104,105]. Duda et al. sought to describe potential biomarkers associated with HCC recurrence following LR or LT. sVEGFR1, VEGF, and VEGF-C levels after LT following LR were associated with prognosis. Specifically, among patients transplanted within the Milan criteria, high VEGF and sVEGFR1 were poor prognostic indicators; however, among patients transplanted outside of the Milan criteria, lower VEGF-C levels were associated with a better prognosis (Figure 2) [106]. In a prospective pilot study of 11 patients, Pommergaard et al. evaluated preoperative blood samples for ctDNA using TruSight Oncology 500 for NGS. Among eight patients who underwent curative HCC resection, only one patient demonstrated a tumor-specific gene mutation in the preoperative sample. However All three patients with advanced HCC had detectable tumor mutations. These data suggest that NGS with ctDNA cannot necessarily be applied to patients with resectable HCC [107].

Due to the high incidence of recurrence following local ablation or resection, Pinyol et al. sought to investigate molecular predictors of HCC prevention with the use of adjuvant sorafenib. In this study, 188 patients were randomized to adjuvant sorafenib (83) or placebo (105); the endpoint of RFS was not achieved. Analyses included gene expression profiling, targeted exome sequencing, IHC, fluorescence in situ hybridization, and immunome. None of the molecular predictors used in the study predicted the benefits or progression following the administration of sorafenib. However, patients in a subgroup that had improved RFS with sorafenib had increased CD4T, B cells, and cytolytic natural killer cells and decreased activated adaptive immune components. Hepatocytic pERK (*p* = 0.012) and microvascular invasion (*p* = 0.017) were also independent prognostic factors [108].

### 3.3. Systemic Therapies

Multiple systemic therapies have been investigated for the treatment of HCC (Figure 3) [109]. Sorafenib had been the mainstay of systemic therapy for HCC for nearly a decade after demonstrating a 3-month survival benefit over placebo. Recently, other first- and second-line therapies have emerged (Table 3) [110,111,112,113,114,115,116,117,118,119,120,121,122,123,124,125]. Response rates remain varied, and PM may help improve outcomes through targeted therapies and improved patient selection (Figure 2).

Atezolizumab (anti-PD-L1) in combination with bevacizumab (anti-VEGF) (atezo/bev) has emerged as new first-line therapy for patients with advanced HCC [126]. Zhu et al. proposed an AFP cutoff ≥ 75% decrease and ≤10% increase from baseline at 6 weeks as a potential biomarker for the efficacy of atezo/bev among patients with HCC, especially HBV-related etiology [127]. Furthermore, Zhu et al. reported an improved clinical response in patients with higher expression of CD271, T-effector signature, and intratumor CD8+ T-cell density using atezo/bev compared with sorafenib or atezolizumab alone. Improved outcomes were demonstrated with atezo/bev versus atezolizumab alone, with increased VEGFR2, regulatory T cells, and myeloid inflammation signatures. A high regulatory T cell-to-effector T cell ratio and expression of GPC3 and AFP were associated with decreased clinical benefit [128]. Additionally, Chon et al. demonstrated that patients treated with atezo/bev with ≥ 30% decrease in AFP and ≥ 50% decrease in des-gamma-carboxy prothrombin had a notably higher objective response rate (42.6% vs. 21.5% and 50.0% vs. 26.2%, respectively *p* < 0.05). Additionally, a neutrophil-to-lymphocyte ratio < 2.5 was associated with a higher objective response rate of 39.0% versus 19.4% (*p* < 0.05). These disease factors could be utilized for further investigations as prognostic indicators of disease progression [129].

The multikinase inhibitor sorafenib had been the mainstay of systemic therapy for HCC for nearly a decade. The multikinase inhibitor lenvatinib demonstrated non-inferiority compared to sorafenib among patients with advanced HCC [114]. Myojin et al. determined that lenvatinib selectively targeted FGF19-expressing tumors, whereas FGF19 inhibition abolished lenvatinib response. Furthermore, FGF19 regulated the secretion of the ST6FAL1 protein in HCC cells, and serum ST6GAL1 correlated with FGF19 expression. In turn, serum ST6FAL1 may potentially be used as a biomarker to identify lenvatinib-susceptible HCC [130]. In a different study by Kim et al., real-time reverse transcription PCR was used to analyze the expression of seven genes (*VEGFR2*, *PDGFRB*, *c-KIT*, *c-RAF*, *EGFR*, *mTOR*, and *FGFR1*) to calculate a treatment benefit score (TBS) of sorafenib in 220 HCC patients. Isolated use of sorafenib resulted in a 0.7–3% response rate in HCC patients, but response rates rose to 15.6% when stratifying patients with actionable genes. *mTOR*, *VEGFR2*, *c-KIT*, and *c-RAF* were the most potent predictors of responders and non-responders [131]. Feng et al. investigated the role of ACSL4 protein expression as a biomarker for sorafenib sensitivity. In an in vivo study, the knockdown of ACSL4 expression by siRNA/sgRNA resulted in greater sorafenib-induced ferroptosis and lipid peroxidation. Therefore, ACSL4 expression was proposed for further investigation as a biomarker to predict the sensitivity of sorafenib in HCC [132]. Rimassa et al. investigated the baseline plasma levels of MET, AXL, VEGFR2, HGF, GAS6, VEGF-A, PIGF, IL-8, EPO, ANG2, IGF-1, VEGF-C, and c-KIT. These investigators noted improved OS and PFS with the use of cabozantinib versus placebo at high and low baseline concentrations of the analyzed biomarkers. Notably, however, low levels of MET, HGF, GAS6, IL-8, and ANG2 and high levels of IGF-1 at baseline demonstrated a potentially favorable biomarker profile in advanced HCC [133].

Regorafenib is a broad kinase inhibitor postulated to prevent angiogenesis, metastasis, proliferation, and immunosuppression through the blockade of VEGFR, PDGFR, KIT, RAF, BRAF, RET, and CSF1R [134]. Teufel et al. examined the expression of plasma proteins and microRNA associated with increased OS among patients with HCC. Low baseline levels of angiopoietin 1, cystatin B, the latency-associated peptide of transforming growth factor beta 1, oxidized low-density lipoprotein receptor 1, and C-C motif chemokine ligand 3 were associated with increased OS with the use of regorafenib (adjusted *p* ≤ 0.05). Furthermore, MIR30A, MIR122, MIR125B, MIR200A, MIR374B, MIR15B, MIR107, MIR320, and MIR645 levels were associated with increased OS related to regorafenib [135].

Camrelizumab is a monoclonal antibody against PD-1. Xia et al. enrolled 18 patients with resectable HCC in an open-label, single-arm trial [136]. Patients received three cycles of neoadjuvant therapy, including three doses of camrelizumab with apatinib for 21 days. The tumor immune microenvironment (TIME), ctDNA, and proteome were examined among responders and non-responders. TIME dendritic cell infiltration was increased in responders versus non-responders, whereas ctDNA revealed a higher positive rate among patients with stage IIb-IIIa disease, and patients with positive ctDNA after surgery had a shorter RFS than individuals with negative ctDNA. Patients with a complete or major pathological response had a higher number of baseline mutations (6 vs. 2.5 mutations, *p* = 0.025) [136].

Nivolumab is an anti-PD1 receptor immune checkpoint inhibitor that prevents the suppression of T cells and immune responses. Sangro et al. reported on the relevance of PD-L1 status with the use of nivolumab among patients with HCC. Using fresh and archival tumor samples from dose-escalation and dose-expansion phases of the CheckMate 040 trial for IHC and RNA sequencing, PD-L1-positive patients treated with nivolumab monotherapy had an increased median OS of 28.1 months versus 16.6 months among patients who were PD-L1-negative (*p* = 0.03) [137].

## 4. Future Perspectives

As the field of PM continues to evolve, a multitude of biomarkers will continue to be identified for early detection, as well as assessment of treatment response and disease surveillance of HCC. In addition, as machine learning becomes more adopted and integrated with clinical and biochemical data, additional predictive models, biomarkers, and targeted therapies will become more prevalent. Artificial-intelligence-based algorithms may aid in rapid data collection and analysis, reducing the time necessary to develop personalized diagnostics and treatments. Machine learning may inform treatment algorithms by accounting for tumor characteristics from data including liquid biopsy, personal risk factors, family risk factors, and environmental exposures. The concept of PM will ideally allow for an individualized approach to cancer treatment, accounting for unique tumor makeup and patient characteristics.

## 5. Conclusions

As HCC continues to increase in incidence globally, personalized medical treatment becomes increasingly relevant [138]. Recognizing obstacles to surveillance, improving diagnostic strategies, and optimizing tailored treatments will be critical to improving outcomes for patients with HCC. Identifying tumor microenvironments through NGS will continue to foster personalized treatment strategies as therapies are tailored to address the molecular heterogeneity of HCC.

## Figures and Tables

**Figure 1 cancers-15-04221-f001:**
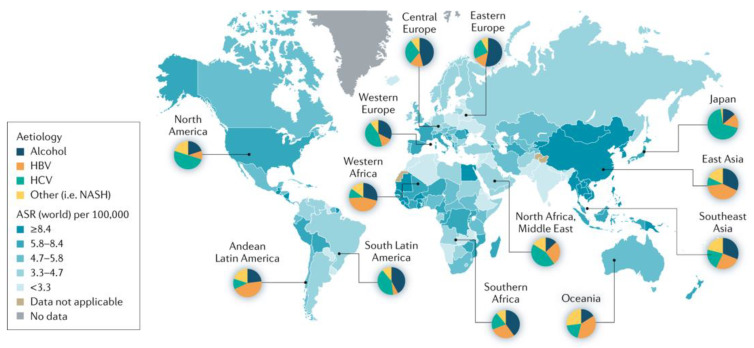
The incidence and major etiological factors involved in hepatocarcinogenesis are depicted in this figure. Abbreviations: ASR—age-standardized incidence rate. Reprinted with permission from Ref. [7]. Global Cancer Observatory, World Health Organization, Estimated age-standardized incidence rates (World) in 2020, liver, both sexes, all ages, Copyright 2020 International Agency for Research on Cancer.

**Figure 2 cancers-15-04221-f002:**
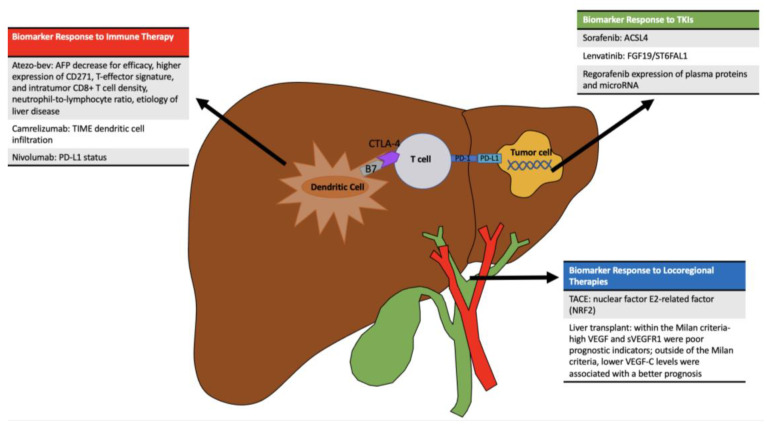
Precision medicine may be used to further increase treatment response through improved patient selection for therapies.

**Figure 3 cancers-15-04221-f003:**
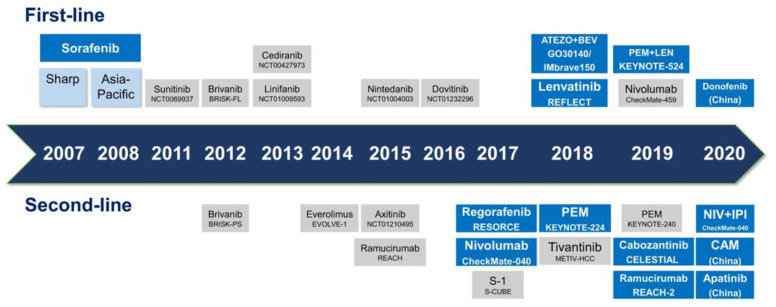
Evolution of targeted agents for HCC. Approved agents are highlighted in blue. Abbreviations: ATEZO—atezolizumab, BEV—bevacizumab, CAM—camrelizumab, LEN—lenvatinib, PEM—pembrolizumab, NIV—nivolumab, IPI—ipilimumab. Reprinted with permission from Ref. [109]. Copyright 2020 Signal Transduction and Targeted Therapy.

**Table 1 cancers-15-04221-t001:** Commonly used techniques in molecular diagnosis. Modified and reprinted with permission from Ref. [16]. Copyright 2022 Surgical Oncology.

Material	Methods	Examples
DNA	Sequencing—Process of determination of the consistent nucleotides of the DNA. First popularized by Fred Sanger, latest techniques called next-generation sequencing (NGS) run millions of these reactions simultaneously, making sequencing faster and cheaper.	Used in exploratory studies and miRNA detection
	DNA probes—Detect specific DNA sequences. They are often tagged with fluorescent markers, which transmit a signal.	Tailored as per need
	DNA microarray—Consists of numerous DNA probes arranged in rows and columns on a small glass surface. Allows for detection of multiple sequences at the same time—so-called ‘high-throughput’ analysis. This allows chip-based detection of multiple variations of the same mutation.	GeneChip®
	Fluorescence in situ hybridization (FISH)—Allows for visualization of the presence and location of specific NDA mutations. These are seen under a fluorescent microscope.	Pancreatobiliary FISH by UroVision
	Polymerase chain reaction (PCR)—Revolutionary technique that produces millions of copies of the desired DNA fragment, which can be detected; nowadays, real-time PCR involves simultaneous amplification and detection, making the entire process faster.	The Cobas® KRAS Mutation Test
	Comparative genome hybridization—Provides an overall picture of chromosomal gains and losses throughout the whole genome of the tumor.	Array-based CGH
	Liquid biopsy—Laboratory testing of bodily fluid samples, including blood or urine, allowing for detection of circulating tumor cells, circulating tumor DNA, cell-free DNA, circulating miRNA, and exosomes. Multiple non-invasive samples may be taken over time, allowing for potential detection, treatment response, and surveillance for disease recurrence.	
RNA	Gene expression testing—These tests study mRNA in the cells to determine activity of different genes.	MammaPrint®, Oncotype DX® Breast
	Reverse transcriptase PCR—Reverse transcriptase is an enzyme that converts RNA into DNA, which is then detected by conventional PCR.	Detection of specific miRNAs
Protein	Immunohistochemistry—Uses antibodies to identify specific proteins. Can provide quantitative and qualitative results.	Pathway Anti-Her2/NEU (4B5) Rabbit Monoclonal Primary Antibody
	Mass spectrometry (MS)—Process of volatilization and ionization of proteins and peptides followed by their detection based on their mass/charge ratio using a mass analyzer. MS may be coupled with liquid or gas chromatography to achieve better separation.	
	Nuclear magnetic resonance (NMR) spectroscopy uses a magnetic field and a radiofrequency pulse to measure organic and some inorganic compounds inside biological samples (as solid tissue or extracted metabolite).	
	Western blot (WB)—Proteins are separated based on molecular weight through gel electrophoresis, then transferred to a band-producing membrane, and the protein of interest is identified through labeled antibodies.	

**Table 2 cancers-15-04221-t002:** Commonly aberrant genes in hepatocellular carcinoma. Reprinted with permission from Ref. [24]. Copyright 2017 Molecular Cancer.

Gene	Aberration Frequency	Pathway	Function	Examples of Potential Targeted Agents
TERT promoter	60%	Telomerase maintenance	Add telomere repeats (TTAFFF) onto chromosome ends, compensating for the erosion of protective telomeric ends that is a normal part of cell division.	
TP53	Mutation: 3–40%; Loss: 2–15%	P53 pathway	Tumor suppressor TP53 gene regulates the expression of VEGF-A. Antiangiogenic agents were correlated with longer PFS in patients harboring PT53-mutant tumors.	Bevacizumab, ramucirumab, sorafenib, and Wee-1 inhibitors
CTNNB1	Mutation: 11–41%	Wnt pathway	Regulates cell adhesion, growth, and differentiation.	BBI608, a potent small molecule inhibitor; PRI-724; and Sulindac
AXIN1	5–19%	Wnt pathway	Regulates cell adhesion, growth, and differentiation.	Small molecular inhibitor XAV939
ARID1A	Mutation: 4–17%	Chromatin remodeling	Transcriptional activation and repression of selected genes via chromatin remodeling.	CDK4/6 inhibitor palbociclib
CDKN2A	Deletion: 7–8%	Cell cycle	Tumor suppressor gene promotes cell cycle arrest in G1 and G2 phases. Suppresses MDM2.	CDK4/6 inhibitor palbociclib
ARID2	Mutation: 5–7%	Chromatin remodeling	Tumor suppressor gene with a role in the transcription, activation, and repression of selected genes.	CDK4/6 inhibitor palbociclib
RPS6KA3	Mutation: 4–7%	Dual-function regulation of MPAK/ERK and mTOR signaling	Mediates stress-induced and mitogenic activation of transcription factors and cellular differentiation, proliferation, and survival.	CDK4/6 inhibitor palbociclib
CCND1	Alterations (focal amplications or deletions): 4.7–7%	P53 pathway cell cycle	Functions as a regulatory subunit of CDK4 or CDK6, the activity of which is required for cell cycle progression.	Palbociclib
FGF3, FGF4, or FGF19	Alterations (focal amplications or deletions): 4–5.6%	FGF pathway	FGF family members possess broad mitogenic and cell survival activities and are operative in tumor growth and invasion, as well as tissue repair.	Brivanib, BIBF 1120, dovitinib, and lenvatinib

**Table 3 cancers-15-04221-t003:** First- and second-line agents for the systemic treatment of HCC with related studies. Abbreviations: AEs—adverse events, CR—clinical response, DC—disease control, HC—hepatocellular carcinoma, MOA—mechanism of action, mo—months, PFS—progression-free survival, TTP—time to progression.

	Setting	Mechanism of Action	Evidence
Atezolizumab + Bevacizumab	Preferred regimen (child class A only) Certain circumstances (child class B only)	Atezolizumab is a monoclonal antibody that binds PD-L1 Bevacizumab is a monoclonal antibody that inhibits angiogenesis by binding to circulating VEGF and interrupting its ability to bind to VEGFR	Atezolizumab + bevacizumab vs. sorafenib: median OS, 19.2 mo vs. 13.4 mo. (95% CI); PFS, 6.9 mo vs. 4.2 mo (95% CI) Atezolizumab + bevacizumab: median OS, 14.9 mo; median PFS, 6.8 mo (95% CI)
Tremelimumab-actl + Durvalumab	Preferred regimen	Tremelimumab is a monoclonal antibody that targets the activity of CTLA-4 Durvalumab is a monoclonal antibody that blocks the interaction of PD-L1 and CD80	Tremelimumab + durvalumab vs. sorafenib: median OS, 16.43 mo vs. 16.56 mo (95% CI)
Sorafenib	Other recommended (child class A or B7 only)	A multikinase inhibitor that works to decrease angiogenesis through inhibition of VEGF receptors, PDGF, and raf kinase	Sorafenib vs. placebo: median OS, 10.7 mo vs. 7.9 (95% CI); TTRP, 5.5 mo vs. 2.8 mo Sorafenib vs. placebo in Asia-Pacific population: median OS, 6.5 mo vs. 4.2 mo (95% CI); PFS, 2.8 mo vs. 1.4 mo
Lenvatinib	Other recommended (child class A only)	A multikinase inhibitor including VEGF, fibroblast growth factor receptor (FGFR), PDGR, KIT, and RET	Lenvatinib vs. sorafenib: median OS, 13.6 mo vs. 12.3 mo (95% CI) Lenvatinib + subsequent anticancer rx. vs. sorafenib + subsequent anticancer rx: median OS, 25.7 mo vs. 22.3 mo (95% CI)
Durvalumab	Other recommended	A monoclonal antibody that blocks the interaction of PD-L1 and CD80	
Pembrolizumab	Other recommended	A monoclonal antibody that binds PD-L1	Monotherapy: median OS, 17 mo (95% CI); median PFS, 4 mo (95% CI) Pembrolizumab vs. placebo in pts previously treated with sorafenib: median OS, 13.9 mo vs. 10.6 mo (95% CI); median PFS, 3.0 mo vs. 2.8 mo (95% CI) Pembrolizumab vs. Placebo in pts previously treated with sorafenib or oxaliplatin-based chemotherapy: median OS, 14.6 mo vs. 13.0 mo (95% CI); median PFS, 2.6 vs. 2.3 mo (95% CI)
Nivolumab	Certain circumstances (child class B only)	A monoclonal antibody that binds PD-L1	Nivolumab vs. sorafenib: median OS, 16.4 mo vs. 14.7 (95% CI)
Nivolumab + Ipilimumab	Certain circumstances (TMB-H tumors)	Nivolumab is a monoclonal antibody that binds PD-L1 Ipilimumab is a monoclonal antibody that binds CTLA-4	Nivolumab + ipilimumab tTMB-H vs. bTMB-H: median OS, 14.5 mo vs. 8.5 mo (95% CI); median PFS, 4.1 mo vs. 2.8 mo (95% CI) Nivolumab + ipilimumab in pts previously treated with sorafenib + N Q2wks (arm A), N + I Q3wks (arm B), N Q3wks + I Q6wks, OR 32% arm A, 27% arm B, and 29% arm C
Second-Line Therapy			
Regorafenib	Child class A only	A multikinase inhibitor including VEGF1/2/3, PDGFR, FGFR1, c-KIT, RAF1, BRAF, and RET.	Regorafenib vs. placebo after sorafenib use: median OS, 10.6 mo vs. 7.8 mo (95% CI)
Cabozantinib	Child class A only	A multikinase inhibitor including tyrosine kinase, c-MET, VEGFR, AXL, and RET	Cabozantinib vs. placebo: median OS, 10.2 mo vs. 8.0 mo (95% CI); median PFS, 5.2 mo vs. 1.9 mo (95% CI)
Lenvatinib	Child class A only	A multikinase inhibitor including VEGF1/2/3, PDGFR, FGFR1/2/3/4, c-KIT, and RET	Lenvatinib vs. sorafenib: median OS, 13.6 mo vs. 12.3 mo (95% CI) Lenvatinib + subsequent anticancer rx. vs. sorafenib + subsequent anticancer rx: median OS, 25.7 mo vs. 22.3 mo (95% CI)
Nivolumab + ipilimumab	Child class A only TMB-H tumors	Nivolumab is a monoclonal antibody that binds PD-L1 Ipilimumab is a monoclonal antibody that binds CTLA-4	Nivolumab + ipilimumab tTMB-H vs. bTMB-H: median OS, 14.5 mo vs. 8.5 mo (95% CI); median PFS, 4.1 mo vs. 2.8 mo (95% CI) Nivolumab + ipilimumab in pts previously treated with sorafenib + N Q2wks (arm A), N + I Q3wks (arm B), N Q3wks + I Q6wks, OR 32% arm A, 27% arm B, and 29% arm C
Pembrolizumab	Child class A only	Immunoglobulin G1 monoclonal antibody that binds to VEGFR and inhibits angiogenesis by decreasing endothelial cell permeability, migration, and proliferation	Monotherapy: median OS, 17 mo (95% CI); median PFS, 4 mo (95% CI) Pembrolizumab vs. placebo in pts previously treated with sorafenib: median OS, 13.9 mo vs. 10.6 mo (95% CI); median PFS, 3.0 mo vs. 2.8 mo (95% CI) Pembrolizumab vs. placebo in pts previously treated with sorafenib or oxaliplatin-based chemotherapy: median OS, 14.6 mo vs. 13.0 mo (95% CI); median PFS, 2.6 vs. 2.3 mo (95% CI)
Ramucirumab	AFP>400 ng/mL and Child class A only	A VEGFR2 antagonist	Ramucirumab vs. placebo in pts previously treated with sorafenib: median OS, 8.5 vs. 7.3 mo (95% CI); median PFS, 3.7 mo vs. 2.8 mo (95% CI)
Nivolumab	Child class B only	A monoclonal antibody that binds PD-L1	Nivolumab vs. sorafenib: median OS, 16.4 mo vs. 14.7 (95% CI)
Dostarlimab-gxly	MSI-H/dMMR tumors	A monoclonal antibody that binds PD-L1	In pts with solid tumors and dMMR/MSI-H: ORR, 87% (95% CI)
Selpercatinib	RET gene-fusion-positive tumors	A kinase inhibitor including wild-type RET and mutated RET isoforms	In pts with RET fusion-positive advanced solid tumors in solid tumors other than non-small cell lung cancer and thyroid cancer: ORR, 43.9% (95% CI)

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
