# Peer review of "Mutational Landscape and Precision Medicine in Hepatocellular Carcinoma"

_cancers, 2023, doi:10.3390/cancers15174221_

Round 1

Reviewer 1 Report

This review is generally very well written, and illustrations are of high quality. The topic is relevant and interesting.

I, however, have some serious concerns:

1.    The section "Methods of Precision Medicine" is textbook material and not mandatory for the understanding of the main topic. I suggest it is moved in modified form to Supplementary materials.

2.    The "Mutational landscape" section is describing single gene variants one after one, their functional importance and possible prognostic and predictive impact. However, a description according to molecular pathways being affected in HCC and their associated genetic variants would be biologically more relevant and easier to read. However, Cancers recently published an excellent review on HCC structured this way: https://doi.org/10.3390/cancers15030817. Moreover, a number of promising and potentially druggable molecular predictive markers are not included, such as tumor mutational burden, MSI, gene fusions and BRCA mutations/BRCAness. Omics signatures are not mentioned.

I suggest that you put the section on possible predictive markers for approved treatments first and the markers for hypothetical treatments last. In fact, most variant found in HCC are today not druggable and the efficacy of drugs suggested, e.g., in Table 2, is not documented. It should be clearly stated that treatment outside clinical protocols is not recommended.

3.    In the section, “Precision Medicine to Guide Therapy”, a review of clinical results of studies of targeted treatments are provided (including also Table 3). This seems somewhat out of scope and, moreover, has been published several times previously.  

In Figure 3 a number of agents are shown that are not approved for HCC - everolimus, sunitinib and others. 

4. Novelty is lacking.

Author Response

  1. The section "Methods of Precision Medicine" is textbook material and not mandatory for the understanding of the main topic. I suggest it is moved in modified form to Supplementary materials.

Thank you for the suggestion- this section was modified and a portion of the information contained in this section was moved accordingly to the introduction per another reviewer’s suggestion.

  1. The "Mutational landscape" section is describing single gene variants one after one, their functional importance and possible prognostic and predictive impact. However, a description according to molecular pathways being affected in HCC and their associated genetic variants would be biologically more relevant and easier to read. However, Cancers recently published an excellent review on HCC structured this way: https://doi.org/10.3390/cancers15030817. Moreover, a number of promising and potentially druggable molecular predictive markers are not included, such as tumor mutational burden, MSI, gene fusions and BRCA mutations/BRCAness. Omics signatures are not mentioned.    

I suggest that you put the section on possible predictive markers for approved treatments first and the markers for hypothetical treatments last. In fact, most variant found in HCC are today not druggable and the efficacy of drugs suggested, e.g., in Table 2, is not documented. It should be clearly stated that treatment outside clinical protocols is not recommended.

The manuscript has been modified and rearranged per your suggestion. A sentence was added to indicate treatment outside of a clinical protocol is not recommended.

  1. In the section, “Precision Medicine to Guide Therapy”, a review of clinical results of studies of targeted treatments are provided (including also Table 3). This seems somewhat out of scope and, moreover, has been published several times previously.  

Thank you very much for your feedback. Part of this section regards how treatment guidelines are informed by precision medicine such as markers or patient populations who may or may not respond to a given therapy.

In Figure 3 a number of agents are shown that are not approved for HCC - everolimus, sunitinib and others. 

Thank you for your feedback. Figure 3 includes drugs that were tested but not approved -- such as the ones you indicated. To be more precise the figure legend was modified.

  1. Novelty is lacking.

This was an invited review article on a topic selected by the editorial team. As such, the novelty of the topic was pre-decided by the editorial team.

Reviewer 2 Report

This is a well-written review regarding the molecular landscape of HCC and the use of PM to assist in determining prognosis, diagnosis, and guidance of therapy. However, some important changes would improve the overall quality of the manuscript to be worthy of publication.

1) None of the manuscript's figures are original, all of them are used with permission. This is not ideal, I suggest removing Figure 2, and adding an original figure regarding the mutational landscape of HCC.

2) The Introduction is way too short, just one paragraph is not enough. At least two "large" paragraphs are required in review articles, with a smaller third paragraph were the aim of the review is explained. Please, modify the manuscript accordingly. 

3) Section 5. Future Directions should be retitled to " Future Perpsectives" as it seems more suitable, and should be increased in content as this part, what the future holds, is very significant for this particular field.

4) Add the following reference in the introduction DOI: 10.3390/cancers15051522

Author Response

1) None of the manuscript's figures are original, all of them are used with permission. This is not ideal, I suggest removing Figure 2, and adding an original figure regarding the mutational landscape of HCC.

Thank you for your feedback. A new figure was constructed and added.  As requested, the previous Figure 2 was replaced.

2) The Introduction is way too short, just one paragraph is not enough. At least two "large" paragraphs are required in review articles, with a smaller third paragraph were the aim of the review is explained. Please, modify the manuscript accordingly. 

Additional information was added to the introduction and as per the other reviewer’s comments. Information from the “methods in precision medicine” section was reduced and moved to the introduction.  

3) Section 5. Future Directions should be retitled to " Future Perpsectives" as it seems more suitable, and should be increased in content as this part, what the future holds, is very significant for this particular field.

Thank you for the suggestion.  As requested, these revisions were made accordingly.

4) Add the following reference in the introduction DOI: 10.3390/cancers15051522

The reference was added as requested.

Reviewer 3 Report

The author reviews the oncogene analyses and gene-based therapeutic effects reported to date in hepatocellular carcinoma. Prospects for precision medicine using biomarkers such as gene mutations are presented.

A large number of reports are well organized and summarized, but I have a few questions and suggestion.

 Major comments

3. Mutational Landscape of HCC: Can you clearly illustrate the site of action of the mutational gene?

Is the expression of the gene mutation different for hepatitis viruses (Type B,C) and non-viruses(NASH)?

 4.3 Systemic Therapies: Please follow the sequence shown in Figure 3.

 Minor comments

Add to abbreviationIncRNA, circRNA, miRNA, mRNA, ncRNA etc. Please check again.

Table 2 and Line 130: PT53 TP53

Author Response

Major Comment

  1. Mutational Landscape of HCC: Can you clearly illustrate the site of action of the mutational gene?

Is the expression of the gene mutation different for hepatitis viruses (Type B,C) and non-viruses(NASH)?

Thank you for the comment. This information was added in the introduction section.

 4.3 Systemic Therapies: Please follow the sequence shown in Figure 3.

The section on systemic therapy was organized to discuss atezo/bev first followed by other therapies. Our goal was not to discuss therapies “in chronological order”.”

 Minor comments

Add to abbreviation:IncRNA, circRNA, miRNA, mRNA, ncRNA etc. Please check again.

These have been checked.

Table 2 and Line 130: PT53 ⇒ TP53

Corrected.  

Thank you for considering our revised manuscript.

Round 2

Reviewer 1 Report

I find the manuscript is now acceptable for publication.

Reviewer 3 Report

Corrected accurately in response to reviewer comments. Illustrations (Fig. 2) have been inserted for easier understanding.

Accept this paper.